# Open-Ended Transmission Coaxial Probes for Sarcopenia Assessment

**DOI:** 10.3390/s22030748

**Published:** 2022-01-19

**Authors:** Paul Meaney, Shireen D. Geimer, Roberta M. diFlorio-Alexander, Robin Augustine, Timothy Raynolds

**Affiliations:** 1Thayer School of Engineering, Dartmouth College, Hanover, NH 03755, USA; shireen.d.geimer@dartmouth.edu (S.D.G.); Timothy.Raynolds@dartmouth.edu (T.R.); 2Geissel School of Medicine, Dartmouth College, Hanover, NH 03755, USA; Roberta.M.diFlorio-Alexander@hitchcock.org; 3Dartmouth-Hitchcock Medical Center, Lebanon, NH 03756, USA; 4Department of Electrical Engineering, Uppsala University, SE-751 05 Uppsala, Sweden; Robin.Augustine@angstrom.uu.se

**Keywords:** sarcopenia, handheld, transmission probe, deep penetration, microwave, broadband

## Abstract

We developed a handheld, side-by-side transmission-based probe for interrogating tissue to diagnose sarcopenia—a condition largely characterized by muscle loss and replacement by fat. While commercial microwave reflection-based probes exist, they can only be used in a lab for a variety of applications. The penetration depth of these probes is only in the order of 0.3 mm, which does not even traverse the skin layer, and minor motion of the coaxial feedlines can completely dismantle the calibration. Our device builds primarily on the transmission-based concept that allows for substantially greater signal penetration depth operating over a very broad bandwidth. Additional features were integrated to further improve the penetration, optimize the geometry for a more focused planar excitation, and juxtapose the coaxial apertures for more controlled interrogation. The larger coaxial apertures further increased the penetration depth while retaining the broadband performance. Three-dimensional printing technology made it possible for the apertures to be compressed into ellipses for interrogation in a near-planar geometry. Finally, fixed side-by-side positioning provided repeatable and reliable performance. The probes were also not susceptible to multipath signal corruption due to the close proximity of the transmitting and receiving apertures. The new concept worked from 100 MHz to over 8 GHz and could sense property changes as deep as 2–3 cm. While the signal changes due to deeper feature aberrations were more subtle than for signals emanating from the skin and subcutaneous fat layers, the large property contrast between muscle and fat is a sarcopenic indication that helps to distinguish even the deepest objects. This device has the potential to provide needed specificity information about the relevant underlying tissue.

## 1. Introduction

Sarcopenia is a muscle disorder distinguished by the loss of muscle mass [1]. It has a direct impact on the strength and function of the combined muscle and skeletal system. It is a progressive disease that is often coupled with obesity, and generally amplifies obesity-based problems in older adults [2,3,4]. In its early stages there is a decrease in muscle quality, while the later stages involve a more pronounced replacement of fibers with fat and different forms of fibrosis [2,5]. These ultimately impact overall operation, including reduced metabolism and impaired mechanical functionality [2,5]. As expected, reduced strength leads to poor life quality, especially affecting the ability to perform daily tasks, and it increases the occurrence of falls and fractures [2,3,4]. Much like osteoporosis, it also leads to increased mortality. Beyond the basic mechanical strength aspects of the entire skeletal system, sarcopenia also impacts the prognosis for multiple conditions and procedures. For instance, the presence of sarcopenia substantially increases the risk of death in women being treated for non-metastatic breast cancer [6,7]. In addition, sarcopenia substantially impacts the outcomes of lung and gastrointestinal cancer treatments [8,9]. The World Health Organization recently recognized sarcopenia as a significant health risk [10]. There is considerable debate as to how to define sarcopenia and subsequently the best ways to diagnose it. Sarcopenia is most often identified individually or by a combination of several metrics, including muscle strength, muscle function, muscle mass, and muscle quality [1]. The former two are essentially measured using mechanical or observational tests, while the latter involve conventional imaging and sensing. Muscle strength for sarcopenia is commonly tested using grip strength or a self-reported SARC-F questionnaire [11]. Patients scoring low on these tests are recommended for further interrogation via imaging and sensing to evaluate muscle mass and/or muscle quality to definitively diagnose sarcopenia. Muscle quantity is measured using conventional imaging (CT, MRI, or US) to quantify muscle thickness or muscle area, or total body lean mass using dual energy x-ray absorptiometry (DXA) [12] or bioelectric impedance analysis (BIA) [13]. Measures of muscle quality use the same modalities to assess tissue architecture. Muscle quality consists of the degree of intramuscular fat (myosteatosis) or fibrosis, which is a secondary manifestation in muscle atrophy. Finally, muscle function tests characterize sarcopenia severity, typically by examining gait speed and other aspects of muscle coordination [11].

To date, the most common populations where sarcopenia is found are in the elderly and in cancer patients. For the former, muscle function is critical and provides insight into patient activity and mobility, which are essential for sustained health [14]. For cancer patients, there is a strong correlation between poor treatment outcomes and muscle mass and quality, which reflect fat infiltration [15,16]. These exams are most often performed using CT, which is considered as the gold standard for assessing sarcopenia in cancer patients. While CT, DXA, and MRI are able to measure muscle quantity and quality, their use is limited because of radiation exposure (CT and DXA), cost, technical expertise, and limited access. Conversely, BIA is portable and uses non-ionizing radiation. However, it is not as accurate as the imaging modalities, does not directly measure muscle quality, does not perform well in obese patients, and cannot be used in patients with electronic device implants. The technique is also plagued by considerable variation, which is attributable to changing hydration levels and exercise status [13].

Studies have shown a good correlation between US and CT measures of muscle size [16]. However, US has predominantly only been used for sarcopenia evaluation in healthy subjects with chronic illness [17]. To date, it has not been used for patients with underlying cancer. However, the potential for ultrasound sarcopenia assessment is high, especially for measuring muscle quality [18,19]. In this situation, fat infiltration of the muscle fibers leads to associated increased echogenicity. US is limited in this setting because the results generally depend on subjective assessment of the tissue echostructure without a calibrated baseline. All of these techniques have advantages and limitations, including overall healthcare costs and access. There is ample opportunity for new approaches to add to and complement the overall armamentarium available to practitioners.

It has been well known for decades that tissue dielectric properties are especially informative regarding health [20]. The most significant factor is water content: fattier tissue with a predominantly low number of polar molecules has low dielectric properties (both permittivity and electrical conductivity) while tissue with higher water content has considerably higher values [21,22]. A clinical example where this may be exploited is edema, where the excess fluid under the skin has high water content which would have very different properties than those for normal skin and subcutaneous fat [23]. However, there are also secondary effects and associated property mixture laws that impact tissue properties and the overall specificity between tissue types. The properties of breast cancer have high contrast with those of normal breast tissue [24,25,26]. Blood has very different properties than those for normal brain tissue, which is particularly useful in imaging for detection of stroke [27]. Normal and osteoporotic bone also have different properties [28,29]. In these applications, the microwave properties could be especially specific for distinguishing healthy tissue from diseased or otherwise compromised tissue. Conceptually, these applications are attractive; however, the barriers to clinical translation are challenging. Full microwave imaging systems are becoming available in limited applications, such as breast cancer imaging and stroke diagnosis; nevertheless, they are large and quite expensive at this stage [30,31,32,33].

Commercial, reflection-based coaxial dielectric probes have been available for several decades. The overall concept is that a signal propagates down a coaxial cable to a clean break in the cable. Nominally, most of the signal is reflected back. However, because a small portion of the fields fringe out into the space beyond the open circuit and eventually reflect back to the coaxes, the composite reflected signal retains information about the material it just interrogated. Sophisticated models have been developed to recover the properties of the object and they usually involve an iterative Newton-like solution [34,35]. This technique is highly reliable and robust commercial solutions are available, e.g., the Dielectric Probe Kit from Keysight Technologies (Santa Clara, CA). However, there are important limitations that inhibit use beyond the standard laboratory setting. Primarily, the penetration depth is particularly shallow. In fact, the depth is nominally a sixth of that of the probe diameter [36,37]. In the context of the Keysight probes, this translates to a depth of roughly 0.3 mm. At best, this is only useful in a superficial application, since it cannot penetrate beyond the skin. Delfin Technologies has exploited this aspect and developed a suite of different diameter probes (as large as 5 cm in diameter) for use in edema diagnosis. The different diameters essentially allow the clinician to characterize the edema at different depths. Equally important, even the slightest perturbation of the cable connecting the probe to the vector network analyzer (VNA) can completely disrupt the calibration [38]. One solution to this is to simply eliminate the cable and implement a rigid (and shorter) connection between the two. This works well but precludes its use in the clinic. The Delfin product avoids this problem simply by operating at a very low frequency (300 MHz) where the wavelengths are quite long and the phase changes due to cable bending are very small. For these reasons, the reflection-based probes are not useful options for clinical work.

A transmission-based probe solves these problems. Previous work by Gaikovich et al. [39,40] demonstrated a broadband, one-dimensional tomography technique to image subsurface inhomogeneities. The operating frequency range was typically in the order of 1.5–7.5 GHz. Simulations in a multi-layered configuration recovered good renderings of the object with somewhat idealized initial distributions [39]. Thorough analytical analysis was performed for the tomography process. It is unclear what type of antenna was envisioned for this application. Later efforts [40] included a pulse-like configuration (correspondingly broadband operation) for a purely homogeneous medium with an associated contrasting target. This method utilized a bow-tie antenna with an operating frequency range of 1.7–7.0 GHz in 801 steps. The properties of the sandy medium and target were not quoted; however, the permittivity and loss for sand were considered low. The results were intriguing because of the depth detection, although they were limited because of noise.

Our first implementation of the transmission concept was for interrogating vertebra during spinal fusion surgery to assess whether the bones were strong enough to withstand the strain from the instrumentation screws [41]. By simply pointing the open-ended coaxial apertures at each other, the small amount of signal leaking from one to the other interrogates the full volume between them. In this case, the primary mechanism for transmission is most likely via the radiation component, since a key observation of that paper was that the transmitting and receiving apertures were within the far field of each other because of their small size. The signal strength is considerably diminished because of the discontinuities of the capacitive coupling at the interfaces of the open-ended coaxes with the material under test (MUT), which are applied uniformly over the band. However, since the spacing distances involved are generally only 1–2 cm, modern VNAs have a sufficient dynamic range to detect the signals. A simple algorithm has been developed to compute the dielectric properties directly from the calibrated measurements [41]. An interesting aspect of this is that while the reflection coefficient at each port is essentially unity (meaning almost all of the signal is reflected back to the source), the associated mismatched impedance (i.e., an open circuit) is applied evenly over a very broad spectrum. In this manner, the transmitted signal transfer function is quite smooth, and the probes can be used from roughly 100 MHz up to about 8 GHz. In addition, operation in transmission mode makes it possible to use without problems from cable motion. In the context of vertebrae probing, the small size of the coaxes and the fact that the surgeon has access to relatively small pedicle holes on both sides of each vertebrae makes this application ideal for the transmission probes.

The operation of these probes in a side-by-side configuration is a direct translation of the aforementioned mutually opposing arrangement. It retains two key features—the broad operating bandwidth and the absence of perturbations from cable bending. It is less obvious what the depth of penetration is because the signals no longer propagate directly across from one aperture to the other. Instead, the signals still fringe out from the transmitting aperture, and some actually spread sideways and are detected by the neighboring receiving aperture. As with the previous configuration, the transmission is not efficient, yet it is still viable over a broad bandwidth, especially using modern, high dynamic range VNAs. Per the needs of this particular application, we added two new features to the design. First, similar to the larger aperture size notion used in the Delfin probes to increase the signal penetration, we increased the diameters. The expectation was that the probes would be able to sense down to roughly 2.5–3 cm, which would be sufficient to get past both the skin and subcutaneous fat layers and begin to interrogate the underlying muscle layer. More interestingly, exploiting 3D printing technology, we were able to fabricate the coaxial apertures as ellipses. We maintained a 50 ohm impedance from the connectors to the outer interfaces. The shape is useful in that it forces the fringing fields that propagate from one aperture to the other to remain in a closer plane than if they had both been circular. This provides a geometric localization advantage when scanning over an area of tissue with a subsurface inclusion. Obviously, fabricating elliptically shaped objects using machining-based technology is a challenge. This is a perfect opportunity for 3D printing technology, which is described in Section 2. One challenge is that there is currently no obvious way to translate the S21 magnitude and phase data into corresponding permittivity and conductivity values. In a real sense, this was not a significant problem, since we were more interested in the signals from the different layers. As will be seen in Section 3, the spectral richness of the data can be exploited to recover information at the different levels. Finally, one of the key features of the new probe is that it can be used in a dynamic process, much like B-mode ultrasound. The ability to manually scan the probe over volumes and at different orientations may provide information that is obscured when used in more limited static modes.

Section 2 describes the probe fabrication process along with the design of the phantom experiments. Section 3 describes the phantom imaging experiments designed to characterize the probe’s ability to assess features at different levels and to locate and distinguish embedded abnormalities. Ultimately, these results indicate that this new concept is sensitive to deep tissue interrogation. It sets the stage for new arrangements of data analysis and even integration with machine learning approaches for clinical investigations.

## 2. Materials and Methods

### 2.1. Transducer Fabrication

A primary requirement for the design was to configure the transmit and receive open-ended coaxes close together such that the signal could propagate from one to the other. We previously designed a related side-by-side coaxial device for interrogating sealed bottles, which was disclosed in Meaney et al. [42]. In that configuration, the internal coaxial transmission lines tapered from small concentric circles (inner and outer conductors) at the connector interface, to larger concentric circles at the probe interface (Figure 1). The circles at the probe end appear distorted and somewhat elongated in the horizontal plane because of how the face of the probe is machined to accommodate the bottle shape. The larger circles make it conducive for deeper signal penetration. Because the circular tapers are classic conical shapes, they can be readily machined using conventional machining technology—both the inner conductor, outer conductor, and the insulator can be machined on a lathe. However, for medical applications, we prefer the plane to be defined by the centers of the coaxes to be confined to a relatively narrow zone even while the overall dimensions of the coaxes need to grow to provide deeper signal penetration. The narrower plane has practical implications during exams in that it improves abnormality localization when scanning from orthogonal directions. This is analogous to a fan beam similar to that used in clinical B-mode ultrasound exams. While the single transmitting ellipse radiates a signal out in all directions, the portion of the fields received by the second ellipse would essentially traverse the shortest path between the two—i.e., along the plane defined by the centers of each. A straightforward way to achieve this is to use elliptical interfaces. The challenge is that these cannot be fabricated easily using machining techniques.

We chose to use 3D metal printing technology to achieve this. A wide range of shapes can be designed using SolidWorks (Dassault Systèmes SolidWorks Corp., Waltham, MA, USA). For instance, it is relatively straightforward to design an object that gradually transitions from a small circle to a larger ellipse, i.e., for both inner and outer conductors. Figure 2a–c show photographs of plastic versions, one as a cross section and the latter two as more complete versions printed in standard polylactic acid (PLA) plastic (note that only one center conductor is shown in Figure 2a). It should be noted that the inner surfaces of the inner and outer conductors were not perfectly smooth because of the 3D printing process. We expect that this may contribute increased signal attenuation. However, since the transmission propagation within the probes is short, the contribution will be minimal and easily accommodated by a high dynamic range VNA. A significant dilemma is how to handle the center conductor and insulator. In this case, we added a simple temporary bridge over the probe interface end that connected the outer housing to the two center conductors and held them in place (Figure 2b). Once fabricated in metal, we used a standard, slow curing resin epoxy as the insulator. The dielectric properties were not exactly that of the more conventional Teflon; however, they were close (relative permittivity of 1.8 for the epoxy versus 2.2 for Teflon at 3 GHz). The epoxy was also significantly more lossy than Teflon; however, the distance the signal travels within the probe is sufficiently short that the overall attenuation was not problematic compared with the large VNA dynamic range. Figure 2d–f show photographs of intermediate versions printed in plastic with the resin immediately after curing, and subsequently after the bridge and excess epoxy were machined off.

After the epoxy cured, there were two additional post processing steps. The first involved machining off the temporary bridge which held the center conductor in place. In this case, some care had to be taken since the 3D-printed aluminum is more brittle than actual aluminum. The final surface was sufficiently smooth for contact with a patient. More intricate machinery is required on the connector end of the housing. First, it needs to have a smooth surface for mating to the N-Type flange connectors (Southwest Microwave 312-04SF, Tempe, AZ, USA). More challenging is that extending from the metal 3D-printed center conductor is a short narrow pin that subsequently inserts into the center of the connector mating surface. Nominally, this needs to be 2.31 mm long and 0.81 mm in diameter. The associated tolerances are too difficult for 3D printing purposes. Instead, a machinist carefully trimmed the pins from a larger 1.65 mm diameter down to the required smaller diameter. This was challenging since the printed metal tends to be brittle. Figure 3a–c show views of the final version. For this particular unit, we deliberately designed it such that the elliptical center conductor was slightly off-center from the center of the outer conductor ellipse. This was done specifically because this unit was part of a set where we intended to further assess the characteristics and determine the optimal dimensions. The final dimensions for this unit are shown in Table 1.

### 2.2. Reference Plane Calibration

In testing the transmission characteristics of devices, components, and materials, it is desirable to calibrate the measurement system such that the measurements only correspond to the object under test—i.e., subtracting out the characteristics of the measurement system as much as possible. For an RF component, this usually means performing a 2-port calibration of the VNA such that the phase reference corresponds to the external planes of the transmitting and receiving coaxial cables. For our transducer, we were primarily interested in the portion of the signal propagating from one open-ended aperture to the other. However, it was not possible to place the faces of the two open-ended coaxes against each other, since they were fixed side-by-side. Therefore, there will always be a need for a baseline measurement that can be performed in some standard material—possibly water or a less polar liquid—depending on the needs of the experiment. One interesting feature we exploited is the fact that all VNA’s are configured with a default calibration with the reference planes set to the output of the connectors on the actual machine. When using relatively short cables between the VNA and the transducer, the amplitudes varied only slightly because of the modest coaxial line loss over our operating frequency range of 100 MHz to 8 GHz. However, the phase change between the VNA default position and the open ends of the transducer varied linearly depending on the cable lengths. The optimal implementation for this scenario was to simply move the phase reference planes to the open-ended interfaces.

This was accomplished utilizing the port extension feature found on most VNAs and had to be performed for both ports. As an example, Figure 4a,b show the phase measurements for the reflected signal at Port 1—3 ft (91.4 cm) long cable—without and with the port extension turned on (4a shows the phase for the full frequency range and 4b shows the phase of a truncated range for closer examination). The case without the extension demonstrated considerable phase wrapping as a function of frequency corresponding to the long transmission line length between the default phase reference plane and the open-end of the transducer exposed to air associated with the cable and internal dimensions of the transducer. The port extension feature allows the technician to arbitrarily add or subtract line length (assuming a 50 ohm characteristic impedance) to the reference plane. The reference plane is positioned at the correct distance once the phase on the screen is close to zero for the entire frequency range, i.e., zero phase corresponds to an open circuit on the Smith chart (same figure but overlayed). In this case, it corresponded to a transmission line length of 1.5063 m or 5.0245 nsec. The phase plot was not perfectly flat, especially at the high frequency end, which was most likely due to some over-moding of the enlarged coaxes in the transducer at the higher frequencies. This process was repeated for both ports. It should be noted that the TE11 mode cutoff frequency for the coaxial dimensions at the end of the probe was roughly 5.3 GHz, implying that we could expect some effect at the higher frequencies [43]. However, this over-moding configuration only exists for roughly one half the length of the tapered transmission lines. In practice, the corruption contributed to the overall propagation is because the propagation velocities of the TEM (our desired mode) and TE11 modes were different [44]. The contribution from this corruption generally accumulates over large distances where it can be quite detrimental. In this case, the distance shared by the two modes was only about 6 cm, for which we observed some unevenness in the port extended phase curve at higher frequencies (Figure 4a).

### 2.3. Phantom Experiments

For the non-homogeneous liquid experiments, the investigations were performed with the phantoms and the active end of the transducer submerged in vegetable oil (Figure 5a). Here, the vegetable oil was used to mimic the properties of the subcutaneous fat layer of the body. In this situation, choosing a liquid material allowed us to explore variations as a function of subcutaneous fat layer thickness, which inevitably varies considerably from patient to patient, especially as a function of body mass index and also as a function of age and gender. For these experiments, the acrylic tank was 49 cm long × 25 cm deep × 30 cm high (Figure 5b). The transducer was supported by a custom, 3D-printed plastic piece which was suspended from a 2-axis positioner, –20 cm horizontal span and 4 cm vertical span (Figure 5c). We placed a thin layer of skin phantom over the face of the transducer while it was suspended in the oil (Figure 5d,e). It was held in place with two thin layers of clear tape (each layer was 0.05 mm thick). The composition of the skin and muscle phantoms was proprietary (Probingon AB, Uppsala, Sweden), and their dielectric properties are presented below in Figure 6a,b along with those for the other materials used in the experiments. We used three variations of the muscle phantom recipe to depict what may arise in a sarcopenic-type scenario, i.e., fat infiltration that would subsequently lower the dielectric properties. Because of the malleable nature of the skin phantom, it was difficult to accurately control the thickness. We report the two different thicknesses used and discuss the impact below. The gel was first mixed and heated in a double boiler and then poured into a rectangular tray for gelling. The trays were 3D printed and were 24 cm long × 14 cm wide × 5 cm deep.

We performed two separate sets of experiments. The first was with a homogeneous muscle mixture, with three separate mixture ratios of the muscle to assess the sensing capability to slight property distinctions (Figure 7a), as would reflect different levels of dispersive fat infiltration [45]. For the second set, we added cylindrical fat equivalent rods to a single mixture at the time of gelling (Figure 7b). The rods are a resin-based solid fabricated according to a recipe by Arthur Guy [46] and the properties are plotted in Figure 11. We performed the tests using three different diameters (1.78 cm, 2.54 cm, and 4.32 cm) as a way to mimic the variations that might be encountered with sarcopenic patients. The rods were supported by custom, 3D-printed holders that were designed such that the maximum vertical position of the rods was just barely under the surface of the gel at the time of curing. Because of the surface tension of the gel in its liquid state, the layer between the top of the rods and the liquid surface ranged from 1–2 mm, and was difficult to control (physical measurements were performed after the experiments). The rods were oriented perpendicularly to the long dimension of the muscle phantom trays and the acrylic tank (Figure 7b).

## 3. Results

### 3.1. Homogeneous Liquid Tests

Figure 8a,b show the S21 magnitude and phase measurements from when the transducer was partially submerged in three different liquids—water, glycerin, and 0.9% saline. The VNA was set to take 201 measurements from 100 kHz to 8.5 GHz with an output power level of 0 dBm. The IF bandwidth was set to 1 kHz, providing a dynamic range of 90 dB assuming that the minimally detectable signal was 10 dB greater than the noise floor (nominally at −100 dBm). In all cases, we used a mean filter which essentially averaged the field values at the given frequency with those from three adjacent frequencies above and below the given frequency. Along with smoothing the desired signal, it had the net effect of smoothing the noise floor signal. In magnitude, the water and saline plots were similar and only extended up to roughly 3 GHz, after which their conductivity became quite large and the associated attenuation made it impossible to detect a signal. It was possible to detect the signals for the glycerin solution all the way up to 8 GHz, since the attenuation was much lower than that for water. The saline magnitude values were predictably lower than those for water across the frequency band because of its higher attenuation. The water plot did exhibit minor perturbations for frequencies below 1 GHz. This is characteristic of multi-path signals, where possibly unwanted signals reflect off the tank walls and are subsequently detected by the transducer. These are normally visible at lower frequencies where the attenuation is lowest. For the associated phase graph, the plots for water and saline were virtually identical. As has been observed in other imaging experiments, at microwave frequencies, the permittivity had a predominant impact on the phase. Since the permittivity of water and saline are virtually identical, it would be reasonable to expect that their phases would be similar. The associated phase values were not meaningful above 3 GHz since the amplitudes were below the noise floor. As expected, the plot for glycerin was considerably different than that for the other two liquids, and could be detected up to 8 GHz because of the lower signal attenuation.

In addition to these measurements, Figure 9a,b show the S21 magnitude and phase plots for the transducer in saline. In this case, the measurements were performed with the transducer held manually and the transducer was removed from the liquid between each measurement. The repeatability was excellent and did not require any form of re-calibration between measurements.

### 3.2. Homogeneous Muscle Phantom Tests

Figure 10a,b show the S21 magnitude and phase plots for the transducer submerged in vegetable oil at different heights above a homogeneous muscle phantom. In this case, the phantom is one of the three whose properties are plotted in Figure 6. These were intended to mimic fattier versions of the muscle, where the fat would be uniformly dispersed within the muscle. The properties for configurations such as these can be estimated using the Maxwell–Fricke mixture law [45]. The heights above the phantom range from 5 mm to 20 mm in steps of 5 mm. The magnitude plots generally exhibited a large lobe spanning from near zero to roughly 4–5 GHz where it is bracketed by a null. Above that, the three thinner spacings exhibited a relatively flat zone from about 5 to 8.5 GHz, while that for the 20 mm spacing was quite uneven over the same span. One of the more interesting features is that the location of the null moved monotonically down in frequency with increasing spacing. The null was deepest for the 10 mm spacing.

Correspondingly, the phase plots exhibited very deep nulls at roughly the same frequencies as those for the magnitude plots. The null for the 5 mm spacing actually transitioned into a phase wrapping. It should be noted that it would have been possible to plot the data without the phase wrapping for the 0.5 cm case. However, we felt this was a useful choice since the phase values for the frequencies above 5 GHz reasonably matched those for the other spacings. We will need to study this phenomenon further. Similar to the magnitude plots, the null locations shifted lower in frequency as a function of increasing spacing. In addition, the depth of the nulls decreased steadily as a function of increasing spacing. These results were for the muscle phantom 1. Figure 10c–f show the same sets of curves except for the muscle phantoms 2 and 3, respectively, which were designed to have incrementally lower properties than those for phantom 1 to mimic a slightly fattier situation. The characteristics were almost identical to those for muscle phantom 1 measurements, indicating that the primary influence was from the varying vegetable oil spacing, whereas the effects from the different muscle phantom recipes was a secondary effect.

### 3.3. Inhomogeneous Muscle Phantom Tests

For the second set of experiments, we used a slightly different recipe for the muscle phantom. Its properties, along with those of the fat equivalent rods, skin, and vegetable oil, are plotted in Figure 11a,b. In these cases, the transducer was scanned horizontally across (over 8 cm) and above the muscle phantom at prescribed heights. The fat equivalent rods were positioned at roughly the mid-points of the scan range and the axis of the transducer was parallel to that of the rods. The rods were supported by custom 3D-printed holders such that their final vertical position was just below the surface of the muscle phantom gel (estimated to be 1–2 mm below the surface). In this case, the skin phantom was only 0.7 mm thick, compared with the 1.5 mm thick phantom used in the previous experiments.

Figure 12a,b show the plots of the S21 magnitude and phase measurements as a function of frequency at the first horizontal position (well away from the fat equivalent rods) for six different heights. Because of the geometry of the tank, the muscle phantom tray, and the transducer support fixture, the closest measurement position was 4.5 mm from the surface of the phantom. The spacing increments were chosen to be smaller when the transducer was closest to the phantom and larger as the separation distance increased, where the measurement change per separation distance naturally diminished. The characteristic spectral shapes were maintained; however, there were distinct variations as the vertical heights changed. The nulls observed in the previous experiments were only modestly visible and had shifted higher in frequency because of the substantially thinner skin layer. For the magnitude, there was a distinct monotonic decrease in magnitude as the spacing between the transducer and muscle phantom increased from zero up to about 4.5 GHz, to the point where there was minimal change between the 15.5 and 20.5 mm cases. In essence, once the gel was far enough away from the transducer, it no longer had an influence. The spacings between curves appeared relatively uniform; however, this was largely because the physical spacings between the measurement positions were also increasing. The phase plots showed similar trends up to about 6 GHz. There was a distinct hump in the data right around 1.7 GHz. The data for the phase exhibited similar qualitative behavior and appeared to be most linear from about 2 GHz to 6 GHz.

For the next experiment, the scan data where the transducer is furthest from the rod—i.e., at horizontal position 0 cm (data from Figure 12)—were used as the reference. In these representations, this means that these reference data (magnitude and phase) were subtracted from the data for all of the other horizontal positions. This normalization isolates the effects due to the presence of the rod without the larger and more pronounced features from the skin and fat. Figure 13a,b show the example S21 magnitude and phase plots at 3 GHz as a function of horizontal position for the parallel orientation and the transducer positioned 4.5 mm above the gel phantom with the 4.32 cm diameter rod. Note that the data point for the horizontal position 0 cm is zero by definition of the normalization process and is not shown. For the phase, there was a large peak in the data near the 4.0 cm horizontal position superimposed on a distinct positive slope in the data. The magnitude plot more closely resembled a sinc function superimposed on a negative slope. On closer examination of the phantom in its tray, there was a slight physical slope in the surface of the muscle phantom gel, creating a height difference of roughly 1 mm from one end of the gel to the other. While this was somewhat disruptive to the overall data analysis, it did emphasize that the probe was remarkably sensitive to this slight perturbation. Note that the normalization did not take the slope into account since the height of the transducer above the gel varies with horizontal position.

Figure 14a,b show the 3 GHz plots of the S21 magnitude and phase as a function of horizontal position for the parallel orientation and for the six different heights of the transducer above the muscle phantom gel. For the magnitude, there was a distinct downward slope as a function of position, and that slope increased with the closeness of the transducer to the gel. The detection of the rod was visible and diminished as a function of the transducer’s position above the gel. There were two somewhat inverted “nulls” to the sides of the main lobe that slightly confounded the detection. The phase example showed a distinct upward slope with respect to horizontal position, and the slope increased as a function of spacing. The detection of the fat rod was obvious and also decreased as a function of transducer-to-gel spacing. It should be noted that the starting vertical position of the transducer aperture was roughly 4.5 mm above the gel surface.

Figure 15a–c show the normalized surface plot for the same phase data in Figure 14b for all frequencies and for the 4.32, 2.54, and 1.78 cm diameter rods, respectively. The fat rod was clearly visible for a relatively large frequency range—roughly 2–4 GHz—after which it monotonically decreased and disappeared by about 6 GHz. The slope was still evident for the 4.32 cm diameter rod case, and seemed to be somewhat uniform across all frequencies except for the embedded slope. The plots were more uniform for the other two phantom cases.

## 4. Discussion

One of the more intriguing aspects of this invention is that the two apertures used to interrogate the materials present capacitively coupled discontinuities to the material under test (MUT). In this situation, this implies that as much as 99% of the signal is reflected back to the signal generator (similar to the receiving aperture, it only receives about 1% of the signal impinging on it). However, the key is that the signal does not need to travel very far and that it operates over a broad bandwidth (modern vector network analyzers can easily measure signals down to −100 dB). For the latter, while the standard circuit analysis implies an open circuit, that open circuit impedance is applied very uniformly across the entire bandwidth. In this way, the fringing transmitted signals can provide broad spectrum characterization of the interrogated target without the unpredictable lobes and nulls that one might observe when operating outside the designated operating bandwidth of a narrower band conventional antenna [47]. Complementary to this is the fact that the signals only need to propagate a few centimeters from one aperture to the other, largely through the target or tissue comprising the space between the two. Because of these features, it has the potential for interrogation into the muscle layers, which could be useful for medical purposes.

There are several important points to make regarding the data. First, it is clear that the skin thickness had a substantial impact on the measurement data. When the thickness was roughly 1.5 mm, the nulls in both the S21 magnitude and phase plots were roughly in the 4–5 GHz range. For the 0.7 mm skin thickness case, the nulls were at the very highest end of the frequency range—roughly 7.5–8.5 GHz. In context, reflection-based open-ended dielectric probes have a well-known property of being quite sensitive very close to the aperture, with this sensitivity rapidly diminishing as a function of distance from the probe face. These data are consistent with that notion; however, the overall effect was predictable, and it appears that there was still ample sensitivity for tissue layers below the skin.

Consistent with the closeness sensitivity impression, two other observations further confirm this assessment. First, the underlying slope of the S21 magnitude and phase measurements shown in Figure 13 and Figure 14 reflected a relatively minor limitation in the phantom fabrication, i.e., there was a slope in the 4.32 cm diameter fat equivalent rod phantom surface amounting a 1 mm height decrease over the span of the 8 cm horizontal transducer scan. Second, it was also clear that the detection of the rods decreased as a function of vertical and horizontal distance from the probe face. Both of these observations confirm that the probe is most sensitive to property deviations near its surface; however, the effect can still extend to a relatively deep level.

With regards to being able to assess tissue below the skin surface, it is paramount to be able to understand the contribution from the subcutaneous fat layer so that its impact can be accounted for. Figure 10 demonstrates a clear progression of the null in both the S21 magnitude and phase plots as a function of frequency. To either side of the magnitude null there were two lobes. Our primary hypothesis is that the lower frequency lobe corresponds mainly to the portion of the signal penetrating the fat and propagating into the muscle and back. This appears quite consistent with the high polar liquid plots shown in Figure 8. The higher frequency lobe more likely corresponds to the portion of the signal that remains primarily within the fat layer—much like the waveguide-like confined signals exploited in the fat channel communications concept developed at Uppsala University [48]. The null between these occurred where the two combined destructively. It is instructive that the magnitude nulls shifted to lower frequencies and became shallower as the separation depth increased. Similarly, the frequency of the phase null shifts was lower with increased separation. These characteristics provide a clear identification of the fat layer which will be utilized in later experiments to allow for the informed characterization of the tissue below the fat layer in patients.

Finally, the plots of the phase for the fat equivalent rod experiments clearly show that the transducer is sensitive to the sarcopenia-like inclusions within the muscle layer. As noted earlier, this is a particularly good application for microwave sensors because of the high dielectric property contrast between fat and muscle, and because the manifestation of sarcopenia will likely occur with relatively large intrusions of fat into the muscle. In addition, this feature occurs over a broad bandwidth, providing the possibility of spectral analysis. We are not aware of comparable probes that can provide signal penetration depth in the order of several centimeters while operating over such a broad bandwidth. The ability to interrogate tissue beneath the subcutaneous fat layer with a handheld device could be opened up to multiple scanning opportunities beyond just sarcopenia.

## 5. Conclusions

We developed a broadband transmission-based probe for deep material and potentially tissue interrogation. The phantom experiments illustrated that we accurately characterized the intervening skin and fat layers, and that information could be derived from heterogeneities in the underlying muscle phantom layers. This is significant in that we do not know of any broadband antennas that can penetrate this deeply. In this case, the depths achieved were significant because we crossed a layer similar to that for subcutaneous fat and still extracted meaningful information from below.

Complementing this achievement is the fact that the probes operate very well in a handheld mode and are not susceptible to debilitating artifacts from cable bending. In most scenarios, this is a challenge for reflection-based probes, albeit they can overcome these associated challenges with continuous routine calibration refreshing. The latter is not always convenient for a real-time clinical evaluation scenario. This probe can be fabricated primarily with modern 3D printing technology and is ideal for operating with ever-shrinking commercial VNAs and control devices.

## Figures and Tables

**Figure 1 sensors-22-00748-f001:**
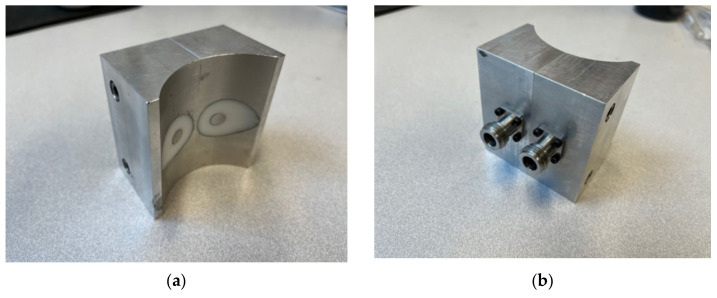
Photographs of a side-by-side transmission probe used for interrogating sealed bottles: (**a**) bottle interface side and (**b**) connector side.

**Figure 2 sensors-22-00748-f002:**
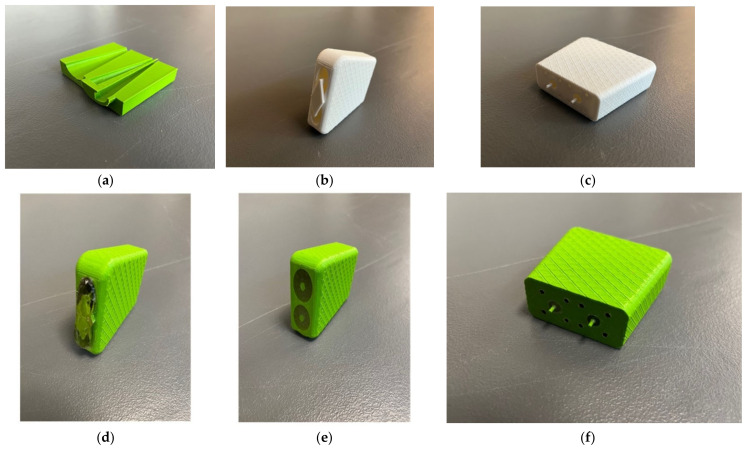
(**a**) 3D-printed cross section of the transducer. (**b**,**c**) Photographs of a preliminary design printed using PLA plastic. (**d**–**f**) Photographs of 3D plastic printed demonstration probes immediately after the resin-based insulator cured and after the bridge and excess epoxy were removed.

**Figure 3 sensors-22-00748-f003:**
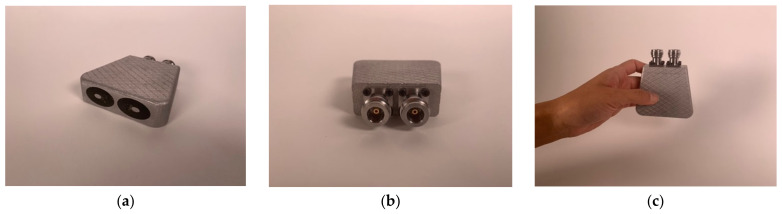
(**a**–**c**) Photographs of the final version.

**Figure 4 sensors-22-00748-f004:**
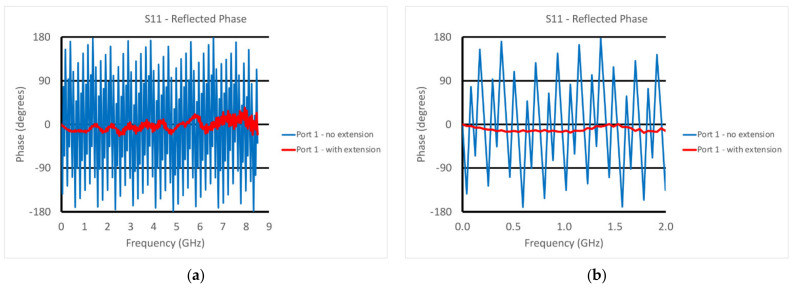
Phase plots for the reflected signal at Port 1 with and without the port extension: (**a**) full frequency range and (**b**) a narrower range (0–2 GHz).

**Figure 5 sensors-22-00748-f005:**
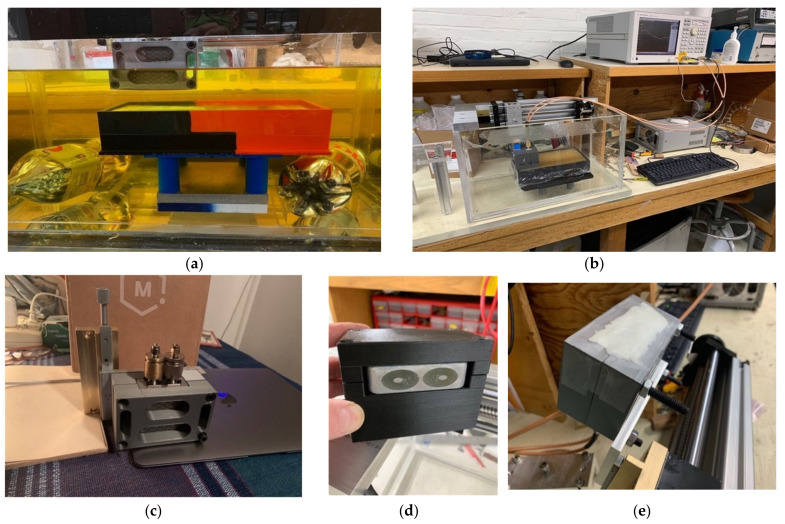
Photographs of the phantom experimental set-up: (**a**) muscle phantom and transducer submerged in vegetable oil, (**b**) transducer suspended by horizontal and vertical positioners in the empty tank, (**c**) custom transducer support structure with vertical positioner, (**d**) transducer mounted in the support structure, and (**e**) transducer in the support structure with the skin layer applied.

**Figure 6 sensors-22-00748-f006:**
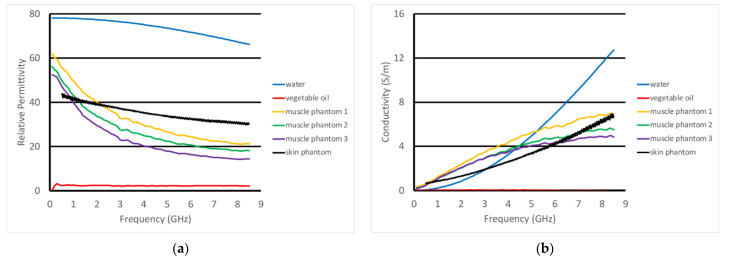
Dielectric properties of the materials used in these experiments: (**a**) relative permittivity and (**b**) conductivity.

**Figure 7 sensors-22-00748-f007:**
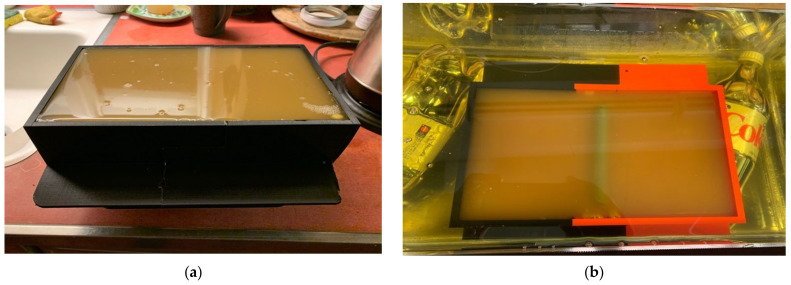
Photographs of the different muscle phantom gels in their 3D-printed trays: (**a**) homogeneous and (**b**) with fat equivalent rod suspended in the gel.

**Figure 8 sensors-22-00748-f008:**
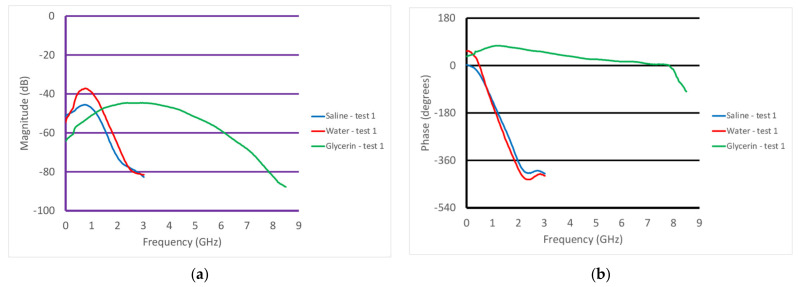
Graphs of the S21 (**a**) magnitude and (**b**) phase as a function of frequency for the cases where the transducer was submerged in different liquids—water, glycerin, and saline.

**Figure 9 sensors-22-00748-f009:**
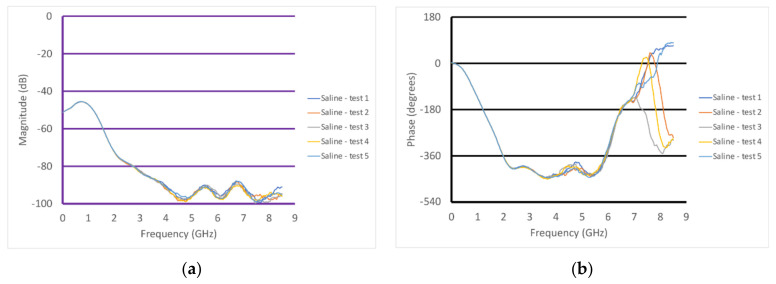
Graphs of the S21 (**a**) magnitude and (**b**) phase as a function of frequency for the case where the transducer was submerged in saline, repeated five times.

**Figure 10 sensors-22-00748-f010:**
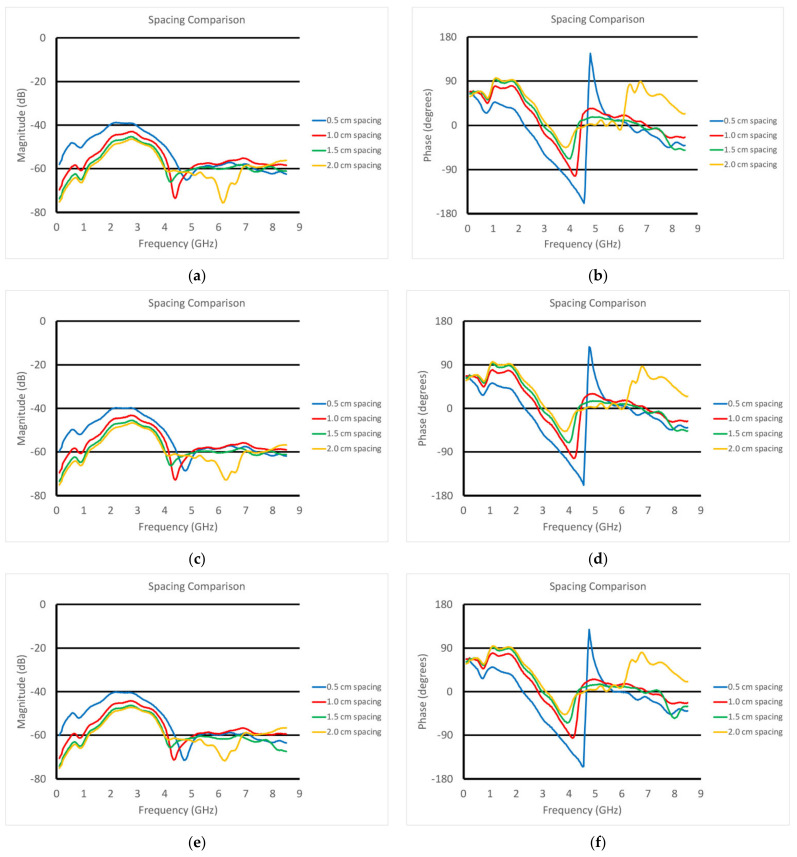
S21 (**a**,**c**,**e**) magnitude and (**b**,**d**,**f**) phase plots as a function of frequency for the transducer submerged in vegetable oil and positioned at different heights above the three muscle phantoms (1—**a** and **b**; 2—**c** and **d**; 3—**e** and **f**).

**Figure 11 sensors-22-00748-f011:**
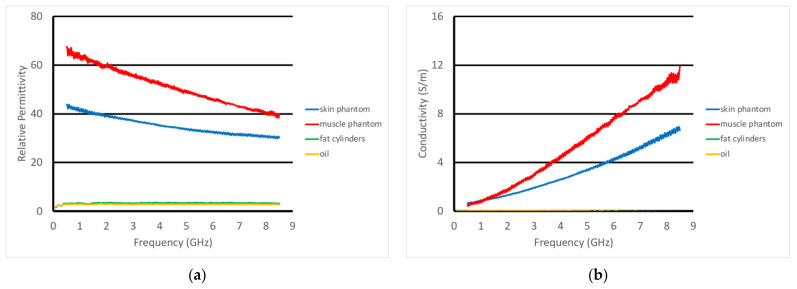
Dielectric properties of the materials used in these experiments: (**a**) relative permittivity and (**b**) conductivity.

**Figure 12 sensors-22-00748-f012:**
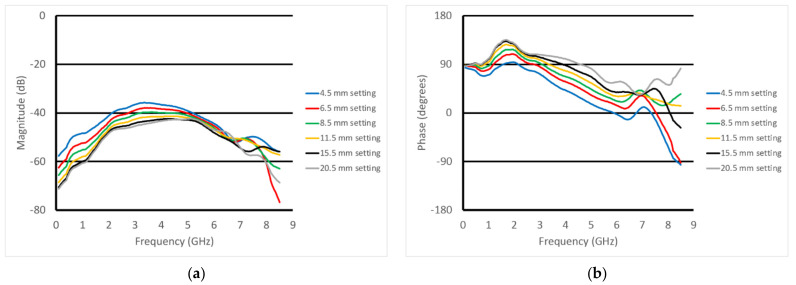
S21 (**a**) magnitude and (**b**) phase plots for the six vertical positions above the muscle phantom at the first horizontal position in the perpendicular orientation.

**Figure 13 sensors-22-00748-f013:**
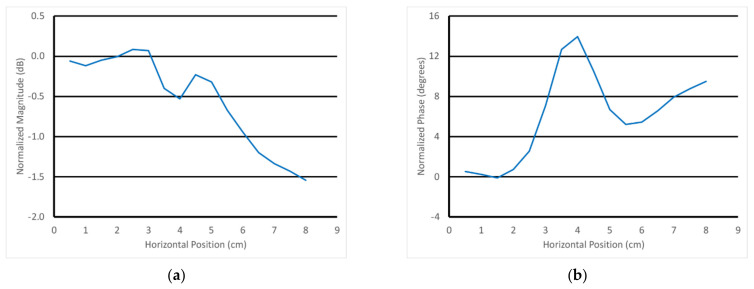
Normalized 3 GHz S21 (**a**) magnitude and (**b**) phase plots as a function of horizontal position for the 4.5 mm vertical position above the muscle phantom with the 4.32 cm diameter rod using the parallel orientation.

**Figure 14 sensors-22-00748-f014:**
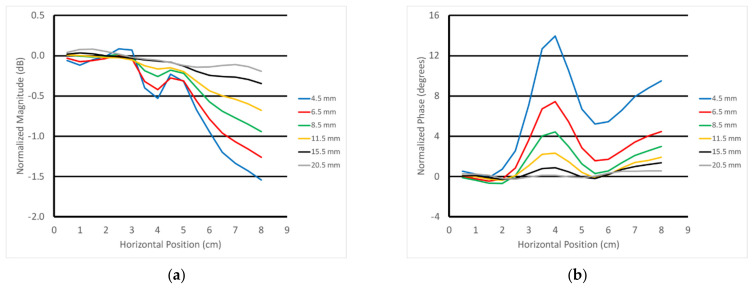
Normalized 3 GHz S21 (**a**) magnitude and (**b**) phase plots as a function of horizontal position for all six vertical positions above the muscle phantom using the parallel orientation.

**Figure 15 sensors-22-00748-f015:**
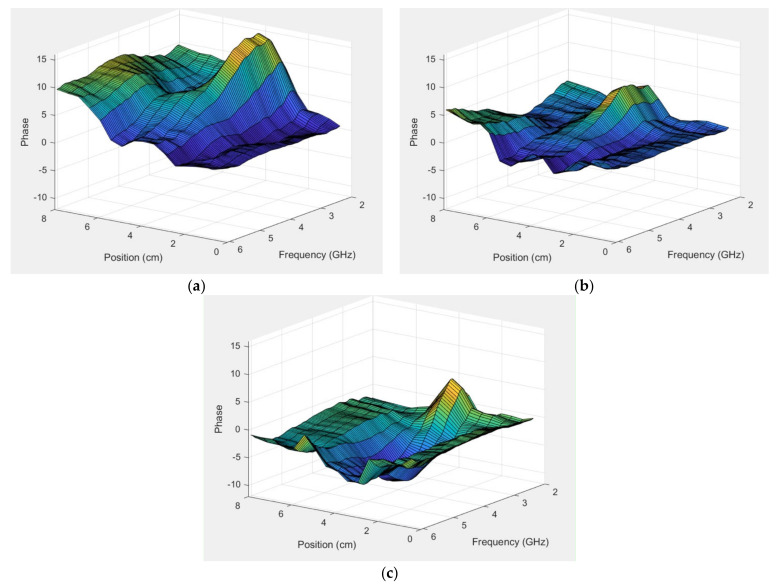
Normalized phase plots as a function of horizontal position for the first vertical position above the muscle phantom and the parallel orientation for all frequencies: (**a**) 4.32, (**b**) 2.54, and (**c**) 1.78 cm diameter.

**Table 1 sensors-22-00748-t001:** Dimensions of the probe transmission lines.

(All Dimensions in Centimeters)	Inner Conductor	Outer Conductor
Connector end diameter	0.17	0.66
Minor (ellipse)	0.38	1.52
Major (ellipse)	0.64	2.54
Length	6.35	
Spacing of connectors	2.03	
Spacing of ellipse centers	2.79	
Off-center spacing of ellipse center conductor	0.19	

## Data Availability

Data will be made available upon request.

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
