# Peer review of "Open-Ended Transmission Coaxial Probes for Sarcopenia Assessment"

_sensors, 2022, doi:10.3390/s22030748_

Round 1

Reviewer 1 Report

The paper describes a proprietary realization of a dielectric sensing probe for a specific application of detecting the fat infiltration into the muscle as a result of sarcopenia.

The realization is technically inventive in terms of transmission-based measurement and proprietary probe realization. Its advantage over the reflection-based probe has been identified in terms of larger sensing depth, justifying further studies and development.

Several remarks are given here, in the order of appearance in the text:

Line 110-111: There exist several Keysight coaxial probes. Does the depth of 0.3 mm refer to all of them, or is it the best case for the whole range of probes? If the penetration depth depends on frequency, does 0.3 mm refer to the upper or the lower frequency limit?

Line 111-112: "it doesn't even penetrate the skin" -> It surely penetrates the skin, it might be better to say that it doesn't penetrate beyond the skin.

Line 127-130: The implementation is no more counterintuitive than the reflection-based probe. Futhermore, the probe is not terminated with an open circuit but with a MUT, just like in the reflection-based implementation. Thinking more about this, the transmission-based implementation might differ from the reflection-based implementation as the reflection-based probing is based mostly on the capacitive load termination (radiation contributes very little to the reflection phenomenon in the working frequency range), while the transmission might be relying more on the radiated portion of the energy. This is actually something that might need at least a bit of discussion. Is the transmission between the transmitting and the receiving probe based on the radiation or the coupling (capacitive or inductive) between the probes?

Line 157-159: Please explain the meaning of "remain in a closer plane". Probably is better to say that it increases coupling between the coplanar transmitting and receiving open ends.

Figure 2 and 3: How did you decide the parameters of the ellipse? Is it a result of an analysis? Please explain, and describe the geometry of the ellipse. Without dimensions, there is no way for other researchers to replicate or fully analyze your work. Allso, what is the role of ellipse excentricity? Please explain.

Line 234-236: The central conductor pin, protruding from the 3D-printed part, was produced with diameter of 1.65 mm, to be narrowed down to 0.81 mm. What is the diameter of the central conductor just inside the 3D-printed part - is it 1.65 mm? If so, is there a discontinuity between the N connector central pin and the central conductor in the 3D-printed part?

Line 259-262: Wouldn't it be more precise to establish the reference plane using the short circuit (metal foil MUT) on the open end, as done when calibrating the open-ended reflection-based probes? Especially since your technique depends on maintaining the 50 Ohm impedance (line 270) along the line within the 3D-printed part, which is probably not fully satisfied concerning possible imperfections of epoxy application and 3D printing tolerance.

Line 270-273: The phase is not expected to be zero throughout the whole frequency range, but only in the lower range and then it should deviate from zero as frequency increases.

Line 273-274: Length of 1.5 meters? How is this related to dimensions of your probe?

Line 274-276: The plot variations can at least partly be a result of geometrical imperfections of the realization of the line geometry and epoxy application (any air bubbles inside the epoxy? does it neatly adhere to the walls of the inner and outer conductor?). The presence of higher modes might be significant, and this should be discussed with respect to the coax line dimensions.

Line 290-292, Figure 5e: How can you be sure that there is no air trapped below the skin phantom layer? Since the skin is not firmly pressed against the open end, it is very possible that there is some air left in between.

Line 312-317: Is this a realistic representation of fat infiltration into the muscle, characteristic for sarcopenia? Or is the fat infiltration usually more diffuse?

Line 314-315: The properties of fat-equivalent rods are actually not shown in Figure 6.

Line 329 & 359: "the magnitude and phase measurements" -> "S21 magnitude and phase measurements" ?

Line 333-336: From these sentences, one could conclude that the actual dynamic range of your VNA was 80 dB. If so, why do you show the line below -80 dB (and phase above 3 GHz for water and saline)? Please state the actual dynamic range of the used VNA to clarify these sentences.

Line 373-374: Is it possible that other spacings also transition into phase wrapping, not observed due to frequency resolution setting? What was the actual frequency resolution/step in these measurements?

Line 383-384: The difference can be described rather as considerable than slight: the relative permittivity increased by 20 with respect to mixtures in 3.2! Please explain the reasoning behind such change. Which mixture is more clinically relevant as a muscle phantom?

Figures 12 and 13: What rod diameter do these figures refer to? Probably the largest diameter of 4.32 cm?

Line 414-415: Please rephrase the sentence to make it more clear. If curves in Figure 13 are normalized to the first position, why the curves don't start at 0 dB? Please define the horizontal position (Figure 13 x-axis) with respect to geometry of the probe and the rod position. Without this, it is not possible to follow the observations described in lines 417-427, and the subsequent observations.

Also consider changing "for this experiment" to "for the next experiment".

Line 455-458: See the comment for line 127-130.

Line 465-466: Why do you reference a narrowband antenna in the context of this application, is there such a realization of this application? If not, this mention is irrelevant.

Line 468-469: This claim ("potential for relatively deep tissue interrogation") is too general, and perhaps too bold. This study already shows some limitations (less/no sensitivity for thick subcutaneous fat layer; large dependance on the skin thickness), therefore, without additional study of the penetration depth, this claim should be more carefully stated. This method seems to provide more depth than the open-ended reflection-based probe, but considering the firm clinical context of this application, it would help to state what the clinically relevant sensing depth is.

General:

- The aimed application is to detect the fat infiltration into the muscle. However, this study, so far, only showed that there is a detectable difference in the S21 magnitude and phase between a homogeneous muscle and the muscle with an inserted fat-like rod. This means that the clinician would have to measure S21 for the homogeneous muscle first. In the clinical scenario, it is questionable whether the clinician will have a solid homogeneous muscle near the fat infiltration, in order to establish the baseline S21. Furthermore, the difference in magnitude is only up to 1.5 dB, and in phase up to ca 14 degrees, and this probably applies to the largest rod diameter (see the comment on Figures 12 and 13). This is very low difference, not much larger than the overall measurement uncertainty of the whole VNA setup. Please comment on the usability of this method from that aspect. Also see the comment on line 312-317.

- In the paper, there is a claim that this method is superior to the reflection-based sensing due to less sensitivity to cable movements. However, this setup only substitutes the coaxial probe with the proprietary two-probe realization in the 3D-printed part. The cables between VNA and the probe are still there, much the same like with the reflection-based probe. As a matter of fact, there are now two cables and two connectors - more critical points sensitive to movements. Please substantiate whether this realization is actually superior from the aspect of sensitivity to cable movements.

Editorial:

Line 49: delete "and"

Line 106: change "i.e." to "e.g."

Line 157: change "if" to "it"

Line 259: change "refence" to reference"
